# Effect of Rotor Geometry on Bending Stiffness Variation

**Risto Viitala \*** , **Tuomas Tiainen** and **Raine Viitala**

Department of Mechanical Engineering, School of Engineering, Aalto University, 00076 Espoo, Finland;
tuomas.tiainen@aalto.fi (T.T.); raine.viitala@aalto.fi (R.V.)
\* Correspondence: risto.viitala@aalto.fi; Tel.: +358-504720016

**Abstract:** Bending stiffness variation (BSV) is a common problem causing vibration in large rotating machinery. BSV describes lateral bending stiffness and its variation as a function of the rotational angle. It has been observed that BSV causes excitation exactly twice per revolution, which leads to vibration problems, especially at half critical speed. BSV is caused by rotor geometry errors if the material is assumed to be homogeneous and linearly elastic. Therefore, the study investigated BSV with harmonic roundness components, which are commonly used in industry to describe the geometry of a rotor. Hence, the results are easily applicable in the industry. The research was conducted primarily by analytical means, but also static simulations and numeric calculations were used. The results clearly showed that when the effect of single harmonic roundness components in rotor cross-sections were observed, only the second component could produce BSV. However, when component pairs were studied, they produced BSV also without the second component. If the second component was included, the profile produced BSV the most aggressively. A generated arbitrary roundness profile, including components 3–50 with random phases and amplitudes, indicated that BSV occurs always twice per revolution despite different components in the profile. The results improve the possibilities of eliminating excessive BSV in the industry, when certain components and component pairs can be avoided.

**Keywords:** asymmetry; bending stiffness variation; half critical vibration; rotor geometry; roundness component



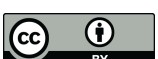

## 1. Introduction

Bending stiffness variation (BSV) is an interesting area of research, as the phenomenon has long been a common problem within industry. Moreover, the prevention of BSV is typically low or non-existent; manufacturing accuracy and tolerances are trusted and separate checks for excessive BSV excitation are not conducted to assure low vibration levels. This, in turn, sometimes leads to serious vibration problems. This research paper tackles the BSV problem, providing novel tools to identify and eliminate BSV already at a manufacturing phase inspecting rotor geometry. This is achieved by investigating BSV from the perspective of the common definition of roundness, which is used to describe the manufacturing quality of round workpieces. For example, machining or casting can produce significant roundness errors to a rotor affecting second moment of area of each cross-section and eventually BSV of a rotor. A correlation between roundness and BSV provides a universal method to estimate BSV caused by geometry errors. The method can be generalized for symmetrical rotors, whose geometry can be measured and defined using roundness components.

Typically, problems related to BSV occur in large rotating machinery operating with slender rotors such as generators and paper machine rolls. Their slenderness increases the sensitivity to vibrations, and large size decreases service and critical speeds. This enables vibration problems at half critical speed, in which BSV excites the rotor twice per revolution [1]. A rotor can end up using this half critical speed due to a dimensioning error, or when it features a wide service speed range or if the foundation stiffness changes

significantly [2]. In this case, resonance and excessive vibration are usually inevitable. BSV can produce excitation only if a rotor is loaded (bent), and thus depending on BSV, a rotor bends differently as a function of its rotating angle. This can be imagined as varying rotor deflection during the slow rotation of a rotor. Usually, a bending load is gravity, but also loads in process and other excitations can bend the rotor and enable BSV excitation [3]. The bending stiffness of a rotor reaches its maximum and minimum twice per revolution due to its principal axes of second moment of area. The excitations of BSV and unbalance must be completely separated. Due to different frequencies and origins, BSV cannot be eliminated by rotor balancing, and therefore, a balanced rotor can still include significant amount of BSV excitation.

Typically, most BSV is caused by rotor geometry errors. Nevertheless, other parameters also affect BSV formation, such as material elasticity and homogeneity. However, these parameters are beyond the scope of the study, and BSV is determined using rotor geometry alone. Typical causes of geometry errors are, for instance, shell thickness variation [4], grooves in the shell [5], welded seams [4,6], keyways, or other manufacturing errors. Sometimes, asymmetric structures are an inevitable consequence of the rotor design, as in the case of two-pole generators and two-bladed propellers, which are prone to BSV. In addition, Juhanko has studied the dynamic geometry of rotors, which obviously also affects BSV due to geometric changes [4]. Further, Darpe et al. have studied cracked rotors, where the cracks were observed to cause similar excitation as BSV [7]. Cracks and studied rotor geometry errors cause BSV similarly by producing nonequivalent principal axes for second moment of area. Hence, the modelling of rotor cracks and geometry errors is very similar when open crack state corresponds to the lowest bending stiffness and closed crack state corresponds to the highest bending stiffness in fixed direction. This paper focuses solely on BSV caused by static geometry errors, which can be described with harmonic roundness components. Thus, cracks or dynamic geometry are outside its scope.

According to the ISO 12181-1 and 2 standards, a roundness profile can be divided into harmonic roundness components that describe the number of lobes in a profile [8,9]. By combining an infinite number of harmonic components, virtually any profile can be described. These harmonic roundness components can be presented with sinusoidal signals whose base frequency is one revolution. The Fourier Transform and its application the Fast Fourier Transform (FFT) are highly related to roundness and its measurement. The FFT is exploited to decompose discrete measured roundness signals into harmonic roundness components that include amplitude and phase information. The FFT can also be used for the inverse purpose of composing roundness profiles from harmonic components. In this case, the FFT is called the Inverse Fast Fourier Transform (IFFT).

Various methods have been developed to measure roundness. The optimal method depends on the size of a workpiece, accuracy demands, and whether the measurement in question is a static or a dynamic measurement. Small workpieces can be measured accurately with a single probe, based on the assumption that there is no run-out in bearings. This assumption is typically made with precision air bearings. Large workpieces are usually measured on their own bearings or separate guiding rolls that disturb the roundness measurement due to the center point movement of a measured rotor. In addition, BSV produces a similar error movement, thereby complicating the measurement. Multi-probe methods, such as the three- [10] and four-point methods [11], can separate the roundness profile and center point movement, and hence their influence on each other can be compensated for. Both methods can also be used in dynamic measurements. Tiainen has conducted comprehensive research on multi-probe methods, their sensitiveness to probe angle errors [12], and probe angle optimization [13], since both methods suffer from mechanical filtering. The four-point method is an extension of the three-point method; it combines a three-point method with a diameter measurement, thus enhancing the distinguishing of even lobes.

With current technology, it is possible to accurately determine rotor geometry, and thus also to identify and eliminate BSV. However, these technologies are typically applied only to simple geometries, such as tubular rotors. Nevertheless, the same methods are

often applicable to the other types of rotors as well. For example, the static geometry of a tubular rotor can be determined by separately measuring an inner and outer surface at multiple cross-sections. Typically, shaft ends are not measured, since the most significant geometry errors are in a rotor body. An inner surface can be measured using the geometry of an outer surface for example with an ultrasonic probe [14]. The procedure can be used for workpieces in which both surfaces are smooth, such as paper machine rolls [4,15,16]. A direct roundness measurement on an inner surface is also possible, but it is challenging due to the lack of space and difficult access especially with slender rotors. The measurement procedure presented above can also be used to estimate the cylindricity of the rotor body. Cylindricity and its measurement have been studied, for example, by Nyberg [17]. The same methodologies can be applied when determining accurate rotor geometry for BSV purposes.

Existing methods are available for decreasing BSV. For instance, Kuosmanen [18] has proposed 3D grinding as a promising method for optimizing rotors under operating conditions and manipulating rotor geometry. Hence, the method is also suitable for the compensation of dynamic geometry change [4] and BSV. Furthermore, other machining methods or methods to manipulate the rotor geometry can also be used, assuming that sufficiently accurate geometric measurement and machining control are available. Some workshops have also manipulated the rotor geometry by applying pressure to the rotor body at different cross-sections, thus compensating for the asymmetry of the rotor. Scientific publications on this method are not available. Other methods, which decrease BSV in tubular rotors, include the turning of the rotor inner surface and the improvement of a steel strip quality [19] and steel strip bending process. The turning of an inner surface reduces its roundness error, thickness variation, and eccentricity. However, if the turning is not implemented for economical or other reasons, the thickness variation of rotor shells can also be reduced by improving the quality of the steel strip or the bending process, which is used in manufacturing of rotor shells. An alternative manufacturing method to the bending is a centrifugal casting, which assures better inner surface roundness.

Previous research has usually discussed BSV as rotor asymmetry or elastic asymmetry that excites second order vibration at half critical speed, secondary critical speed, or gravity critical speed. These studies are typically conducted as simulations or analytical models which investigate how BSV affects dynamic rotor behavior. Bishop and Parkinson [1] were inspired by the BSV problems of large two pole alternator rotors whose rotor design was prone to asymmetry. In their research, a modal steady-state model was developed that was not restricted simply to Jeffcott's theory of rotor dynamics; the model included a uniform asymmetric rotor with a constant rotor profile. As a result, rotor behavior with a certain asymmetry could be investigated, and natural frequencies could be estimated in addition to the twice-per-revolution vibration at half critical speed. In turn, Brosens and Crandall [20] and Yamamoto and Ota [21] studied regions rendered unstable by rotor asymmetry. Their results indicated that unstable regions near the first critical speed and at higher speeds could be manipulated with the degree of asymmetry and damping properties. Messal and Brontron [3] conducted experimental and analytical research on a vertical asymmetric rotor that eliminated the effect of gravity on lateral vibrations, which typically enables twice-per-revolution excitation resulting from asymmetry. However, their results indicated that similar excitation still existed, even though gravitation had been eliminated. Messal and Bronton [3] estimated that parametric excitation and bearing misalignment caused the BSV excitation in their experiments, but also unbalance is observed to increase the twice-per-revolution excitation. Thus, there can also be other excitations that induce half critical vibration directly or enable BSV excitation due to rotor bending.

Previous experimental research concerning BSV is sparse, since the measurement and manipulation of rotor geometry and BSV is demanding and laborious. BSV has been investigated experimentally but also with analytical and simulation methods in the author's still unpublished study. In this earlier study, the first links between BSV and rotor geometry were established when the effects of single harmonic roundness components

and lateral load on developed BSV excitation were studied. The research indicated that only the first and second components can cause BSV in a tubular rotor. In a solid profile and rotor, the only affecting component is the second component. Experimental tests and simulations indicated the effect of component amplitude on BSV excitations and rotor vibrations. Expectedly, larger component amplitude and lateral load increased developed BSV excitation, and thus rotor vibration. This present paper extends the earlier BSV research to concern more realistic rotor cross-sections containing several components in their roundness profiles.

In modern industry, analytical models and experimental tests have frequently been replaced by the Finite Element Method (FEM) simulations, of course depending on the complexity of the application. Analytical models still have advantages in simple cases, in which the same results can be achieved also with lighter models. However, FEM has proven to be more accurate and flexible when estimating the dynamics of more complex rotors. FEM allows rotor geometry, as asymmetry, to be analyzed more accurately, within the limits of computing speed, which determines the reasonable meshing resolution. Kang et al. [22] have generalized a FEM formulation to present the effects of both the deviatoric inertia and stiffness due to asymmetry of a flexible shaft and disk. The model, built with 1D elements, is based on Timoshenko beam theory and takes account of gyroscopic moment, rotary inertia, shear deformation, and asymmetry. In addition, Heikkinen et al. [15] have also studied the behavior of an asymmetric rotor based on Timoshenko beam theory. In their model, the rotor, its asymmetry, and its uneven mass distribution were modeled using thickness variation data from the actual test rotor, which resulted in a realistic description. Even more accurate FEM models can be achieved when moving from 1D elements to 3D elements. For instance, Meng et al. [23] have produced a good example of a 3D FEM, which is used to estimate the dynamic response of an asymmetric rotor. 3D elements can describe more complex geometries, such as flexible blades and disks with dynamic couplings. In addition, 3D CAD models can be directly exploited in the analysis. However, 3D FEM demands far more processing power compared to 1D FEM, which has limited its adoption. Therefore, the final model is often some kind of compromise.

The research investigates the effect of rotor geometry on BSV. Rotor geometry was described using harmonic roundness components, and different cases were created to correspond to realistic rotor geometries. Hence, rotor geometry was described separately with single components, component pairs, and component combinations, and their effect on BSV was studied. The investigation of single components indicated that only the second component (ovality) produced BSV, and the observed BSV occurred twice per revolution. Investigation of component pairs showed that certain pairs were able to produce BSV twice per revolution as well, even though neither were the second component. This indicates that the superposition principle does not apply in the BSV problem. When comparing the effects of component pairs and the second component on BSV, the second component was observed to produce significantly larger amplitudes. Lastly, an arbitrary roundness profile was investigated, and results indicated that even if the profile included various components with randomly generated amplitudes and phases, the produced BSV occurred twice per revolution. However, the BSV amplitude was lower than with the second component alone or with any BSV-producing component pair, and it seems that a component mix only decreases BSV. The effect of the component phase on BSV was studied with a static load simulation. The numerical simulations indicate that the influence of components on overall BSV in a rotor can be compensated for by positioning them at different cross-sections along the rotor. The results of the paper can be used to identify BSV and eliminate BSV in rotors as such. At the minimum, the measurement of rotor geometry must be conducted, and geometry described with roundness components. Decreased BSV excitation increase the operation reliability of rotating machines and prevent the formation of insurmountable problems on-site.

## 2. Methods

This section presents the investigation methods used to produce the results of this study. The research applied analytical, numerical, and simulation methods.

### 2.1. Bending Stiffness Variation of an Asymmetric Rotor Described Using Roundness Components

The asymmetry of a rotor can be described in terms of harmonic roundness components. Here, rotor asymmetry can be unwanted manufacturing error on a circular rotor. The typical manufacturing errors are harmonic, and thus exploiting harmonic roundness components is possible. However, describing manufacturing errors and rotor geometry with roundness components is only the estimate of actual geometry. The ability of roundness components to describe rotor geometry and manufacturing errors weakens if the nature of geometry errors is non-harmonic such as cracks in a rotor.

Roundness components refer to undulations, their amplitudes, and phases on the circular profile. The concept of roundness and its components are introduced in ISO 12181-1 and ISO 12181-2 [8,9]. Roundness profiles can be used to study BSV when calculating second moment of area for a described profile. The second moment of area describes the resistance of the profile shape to bending with respect to its neutral axis. Hence, the bending stiffness of a rotor can be studied exploiting only second moment of area with certain conditions: material must be linearly elastic and homogeneous. In the other words, Young's modulus must be constant.

The Fourier series is strongly related to roundness components, as it can be used to decompose the finite length signal into sinusoidal components. The rotor cross-sectional profile can be considered a finite and continuous signal, and thus it can be described with sinusoidal components. The fundamental sinusoid is termed the first harmonic of the signal (eccentricity), while the sinusoid that is half of the length of the fundamental sinusoid is known as the second harmonic (ovality); in turn, the third harmonic (triangularity) is one third of the length of the fundamental sinusoid, and so on and so forth. Figure 1 presents the roundness components with corresponding sinusoidal components in polar coordinates.

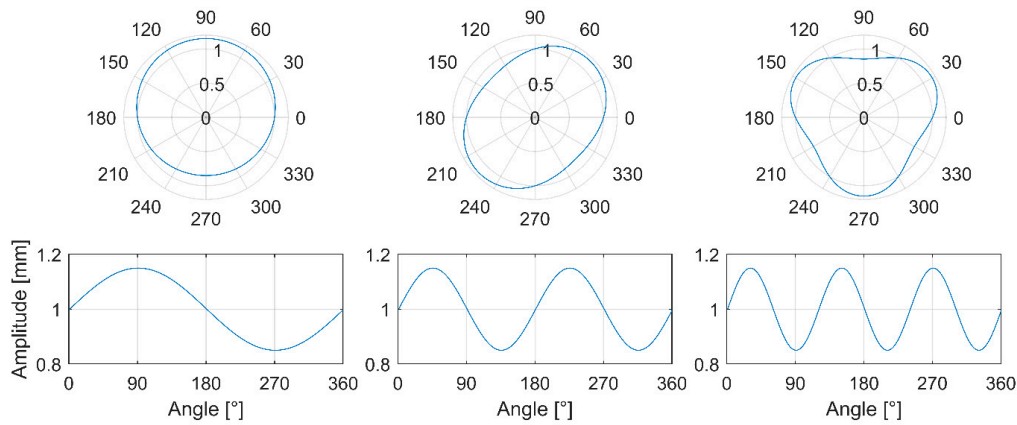

**Figure 1.** First three sinusoidal roundness components plotted in polar and cartesian coordinates.

The Fast Fourier Transform (FFT), which is based on the Discrete Fourier Transform (DFT), is commonly applied to discrete roundness measurements to transform time domain data into a frequency domain. In practice, the results of the FFT analysis consist of a finite number of different components with certain amplitudes and phases depending on the resolution of a data acquisition. Once separated into its roundness components, it is possible to recompose the decomposed roundness profile.

### 2.1.1. Bending Stiffness Variation of a Rotor Described with a Single Roundness Component

Roundness profiles contain a virtually infinite number of different components, but frequently the profile is dominated by only one or a small number of components. There-

fore, the investigation begins by inspecting profiles consisting of just a single roundness component. The effect of a single roundness component on BSV has also been studied in the author's still unpublished paper. BSV and the second moment of area of a roundness profile can be calculated using polar coordinates. Equation (1) below presents the formula of the second moment of area in cartesian coordinates and Equations (2) and (3) in polar coordinates.

$$I_x = \iint y^2 dA,$$ (1)

$$I_x = \int_0^{2\pi} \int_0^r r^3 \sin^2(\theta) dr d\theta,$$ (2)

$$I_x = \int_0^{2\pi} \frac{r^4 \sin^2(\theta)}{4} d\theta,$$ (3)

where $y$ presents the position of a calculated point in $y$-axis, $r$ is the radius of a profile, and $\theta$ is the angle between $x$-axis and a calculated point. Transition between two different coordinate systems is illustrated in Figure 2.

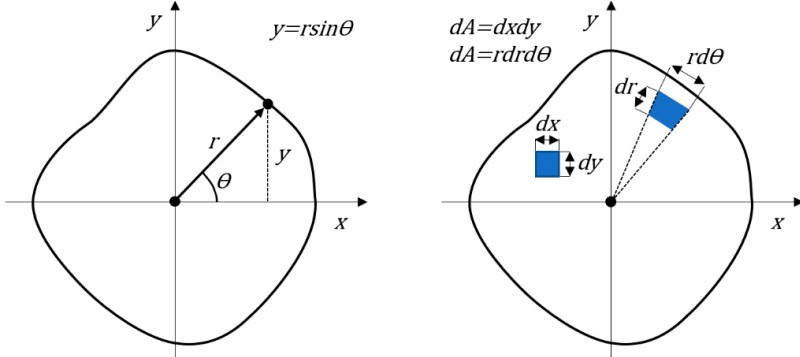

**Figure 2.** Transition between two different coordinate systems. On the left-hand side: $y$-axis position noted in cartesian and polar coordinates. On the right-hand side: integral partitions in cartesian and polar coordinates.

The formula can be applied to ideally circular profiles with a constant radius. However, if roundness components are intended to be included in the formula, the radius must be varied. This can be achieved by substituting the roundness component in sinusoidal form for the variable $r$:

$$r = R + Asin(k(\theta + \phi)),$$ (4)

where $R$ is the constant radius of a circle, $A$ is the amplitude of undulation, $k$ is the number of undulations per revolution, and $\phi$ is the phase angle.

Hence, when Equation (4) is substituted for Equation (3), the effect of each roundness component on the second moment of area and BSV can be separately studied by varying the constant $k$. Figure 3 illustrates rotors in which the roundness profile is constant along the rotor and consists of only a single component. The formulas of this section apply only to this kind of rotor.

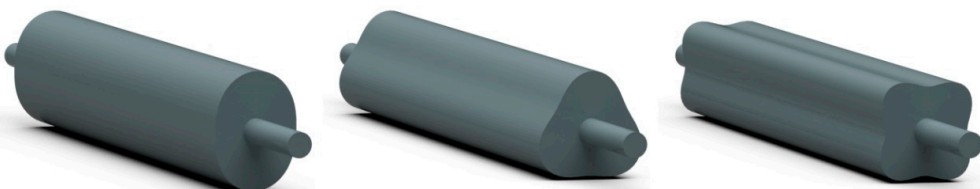

**Figure 3.** Rotors with a single roundness component. From left to right: second component, third component, and fourth component.

2.1.2. Bending Stiffness Variation of a Rotor Described with a Roundness Component Pair

Roundness profiles can also include another significant roundness component. The effect of roundness component pair on the second moment of area and BSV can be investigated when Equation (4) is expanded to include another sinusoidal roundness component:

$$r = R + A_1 sin(k_1(\theta + \phi)) + A_2 sin(k_2(\theta + \phi)), \tag{5}$$

where $A_1$ and $A_2$ are the amplitudes of the roundness components, and $k_1$ and $k_2$ indicate the number of the components.

When substituting Equation (5) for Equation (3), the second moment of area of the profile including two roundness components can be solved as a function of phase angle $\phi$, and thus BSV can be investigated. Figure 4 illustrates rotors in which the roundness profile is constant along the rotor and consists of component pairs. The formulas of this section apply only to this kind of rotor.

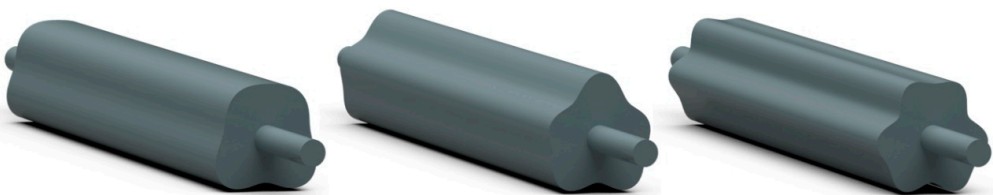

**Figure 4.** Rotors with harmonic roundness component pairs. On the left: components 3 and 4, in the middle: components 3 and 5, on the right: components 3 and 6.

2.1.3. Bending Stiffness Variation of Rotors Described with an Arbitrary Roundness Profile

The effect of an arbitrary roundness profile on the BSV of a rotor can be investigated by generating a roundness profile that includes dozens of different components. However, arbitrary profiles cannot be investigated analytically, as in previous Sections 2.1.1 and 2.1.2, since the symbolic solution would be too large to handle and express rationally. Nevertheless, arbitrary roundness profiles can be investigated numerically when the second moment of area of an arbitrary profile is interpreted as a function of the rotation angle. Figure 5 illustrates a rotor with an arbitrary roundness profile.

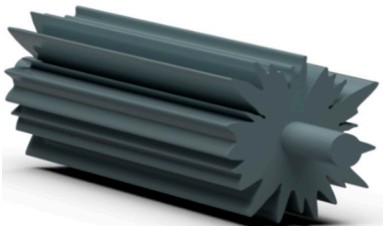

**Figure 5.** A rotor depicted with randomly generated harmonic roundness components.

An arbitrary roundness profile with numerous roundness components can be created using the Inverse Fast Fourier Transform (IFFT). The IFFT transforms the given phase and amplitude of each roundness component back to a discrete time domain signal, which is,

in this case, the generated roundness profile. Hence, the process is completely the reverse of that performed in the FFT. After this, the second moment of area can be calculated using the roundness profile created and applying numerical integration. In this study, MATLAB was used to create the profiles and calculate the second moment of areas as a function of the rotating angle.

### 2.2. Bending Stiffness Variation of an Asymmetric Rotor with a Varying Cross-Section

The BSV of a rotor is dependent on the phase and amplitude of the harmonic roundness components in the roundness profiles along the rotor. Virtually all manufactured rotors have a roundness profile that includes a large number of different harmonic components and varies along the rotor. However, the lowest roundness components are typically the most significant in terms of the occurrence of BSV. In addition, these components mainly occur at the same phase along the rotor due to typical manufacturing errors.

The previous sections discussed roundness profiles that remain constant along a rotor. Hence, the study of BSV was straightforward, as only one roundness profile and its second moment of area needed to be interpreted. When a rotor possesses a varying roundness profile, the effect of the component phase on the final BSV is of particular interest, since the phase of the components can be expected to affect overall bending and thus BSV. Therefore, some workshops have attempted to compensate for BSV by reversing the roundness profiles in certain cross-sections.

The effect of the amplitude of the roundness profile components and the profile location along a rotor is obvious; the amplitudes of components directly influence the BSV amplitude (if they produce BSV), and the location of the profile along the rotor produces different effects on overall bending due to larger moments at the middle of the rotor. After this, the only parameter of interest is the phase of the roundness profile along the rotor.

The effect of the phase of the roundness profile can be studied by removing other parameters. This is achieved by creating a rotor that includes only one BSV-producing profile. In this study, it is generated by adding only the amplitude of the second component to the roundness profile and rotating the profile with different angles along the rotor. The effect of the component phase on BSV can be observed when interpreting the bending at the middle cross-section of the rotor when the rotation of the component is varied. Four different rotors were studied. These rotors are illustrated below in Figure 6.

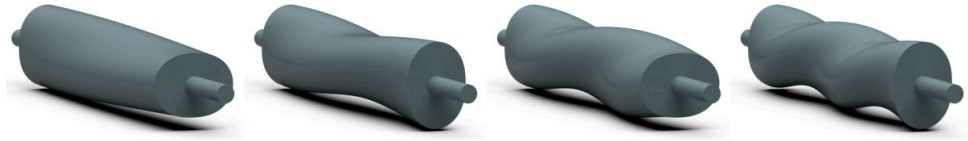

**Figure 6.** Rotors with a single roundness component that includes only the second component. The phase of the component varies continuously along the rotor. From left to right, the component rotates 90°, 180°, 270°, and 360°. The phase shift is distributed evenly along the rotor.

The bending of the middle cross-section can be measured and interpreted with a simulation in which the rotor is statically loaded. In the present study, the static loading was performed using the Finite Element Method (FEM). Similar rotor models to those presented in Figure 6 were constructed, in which the BSV-producing profile (oval profile) was evenly distributed along the rotor as a function of the rotation angle. The rotor end shafts were excluded from the simulation model, since, typically, most BSV is produced in the rotor body. The modeled rotor bodies were 1 m long with a 100 mm radius. Ovality was created by adding the second component to the roundness profile. In these rotors, the second component had an amplitude of 15 mm. With these parameters, the BSV of the oval roundness profile was approximately 57%.

The FEM models of the rotors were created with Siemens NX. An oval roundness profile was used to extrude the rotor bodies, which were thus solid. As demonstrated in Figure 6, the rotor bodies were twisted around their rotation axis so that the effect of

the varying phase on BSV could be studied. The rotor models were divided into 25 mm tetrahedral 3D elements to provide sufficient accuracy. The rotors were then loaded with gravity at 36 different angles at 10-degree increments. The rotor ends were laterally constrained, and, additionally, axially constrained only at one end. The displacement of the center point of the rotor middle cross-section was acquired from each loading case. Figure 7 illustrates the described FEM simulation.

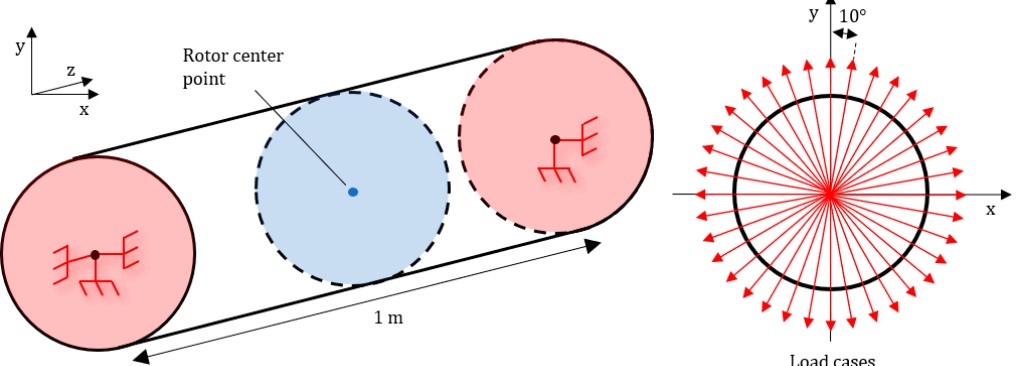

**Figure 7.** Constraints and loads of the FEM simulation. On the left-hand side are presented the positions and directions of simulation constraints. The constraints affected the whole surface at the rotor ends. The displacement of rotor center point was measured. On the right-hand side are presented 36 simulated loading cases using gravity.

## 3. Results and Discussion

This section includes the results produced using the methods introduced in the previous section. The section is structured similarly to the methods section. The discussion follows the results in each subsection.

### 3.1. Effect of a Single Roundness Component on Bending Stiffness Variation

The effect of a single harmonic roundness component on the second moment of area and BSV could be calculated and studied using Equation (3). These equations allow the analytical solutions for each harmonic roundness component to be solved as a function of phase angle $\phi$, which describes the rotation of the profile. Equations (6)–(8) below present the analytical solutions for harmonic components 2–4.

$$k = 2, \qquad I_x(\phi) = \frac{\pi}{4}R^4 - \frac{\pi A \sin(2\phi)}{2}R^3 + \frac{3\pi A^2}{4}R^2 - \frac{3\pi A^3 \sin(2\phi)}{8}R + \frac{3\pi A^4}{32} \qquad (6)$$

$$k = 3, \qquad I_x(\phi) = \frac{\pi}{4}R^4 + \frac{3\pi A^2}{4}R^2 + \frac{3\pi A^4}{32} \qquad (7)$$

$$k = 4, \qquad I_x(\phi) = \frac{\pi}{4}R^4 + \frac{3\pi A^2}{4}R^2 + \frac{3\pi A^4}{32} \qquad (8)$$

Equations (6)–(8) show that only the second harmonic component ($k = 2$) produces BSV, since the solution includes phase angle $\phi$, and thus produces variation in bending stiffness when the profile rotates. BSV is obviously produced twice per revolution, since phase angle $\phi$ is multiplied by two inside the trigonometric function that yields two periods at range $0 \dots 2\pi$ (one revolution). Other components ($k = 3$, $k = 4$) produce constant bending stiffness despite the phase angle or rotation of the profile. The same trend also continues with the higher components (no BSV) if more components are calculated. In Appendix A, is more detailed discussion about BSV-producing roundness components.

The first component, which describes the eccentricity of the profile (illustrated in Figure 1), does not affect the shape of the profile. Thus, it cannot affect BSV, and therefore it is not presented. However, this applies only to solid profiles; in the tubular rotor case, if the first component describes the eccentricity of an inner or outer profile, it affects BSV.

The results indicate that if the cross-sections along a rotor consist mainly of one significant roundness component, only the second component produces BSV in the rotor

and BSV occurs twice per revolution. Typically, the roundness profile also consists of other components, but frequently only a few low components are significant. The case of one dominant component is possible, perhaps even common, in industrial rotors.

### 3.2. Effect of Roundness Component Pairs on Bending Stiffness Variation

The effect of roundness component pairs on BSV can be analytically solved by applying the formulas introduced in Sections 2.1.1 and 2.1.2. For example, the second moment of areas for the profiles including component pairs 3 and 4, 3 and 5, and 3 and 6 can be solved as follows:

$$
\begin{aligned}
k_1 &= 3,\\
k_2 &= 4,
\end{aligned}
\qquad
I_x(\phi) = \tfrac{\pi}{4}R^4 + \frac{\pi\left(24A_1^2+24A_2^2\right)}{32}R^2 + \frac{3\pi A_1^2 A_2 (\sin 2\phi)}{8}R + \frac{\pi\left(3A_1^4+3A_2^4+12A_1^2 A_2^2+3A_1^2 A_2^2\left(2\sin^2\phi-1\right)\right)}{32}
\tag{9}
$$

$$
\begin{aligned}
k_1 &= 3,\\
k_2 &= 5,
\end{aligned}
\quad
I_x(\phi) = \tfrac{\pi}{4}R^4 + \frac{\pi\left(24A_1^2+24A_2^2+24A_1 A_2\left(2\sin^2\phi-1\right)\right)}{32}R^2 + \frac{\pi\left(3A_1^4+3A_2^4+12A_1^2 A_2^2+6A_1 A_2^3\left(2\sin^2\phi-1\right)+6A_1^3 A_2\left(2\sin^2\phi-1\right)\right)}{32}
\tag{10}
$$

$$
\begin{aligned}
k_1 &= 3,\\
k_2 &= 6,
\end{aligned}
\qquad
I_x(\phi) = \frac{\pi}{4}R^4 + \frac{\pi\left(24A_1^2 + 24A_2^2\right)}{32}R^2 + \frac{\pi\left(3A_1^4 + 12A_1^2 A_2^2 + 3A_2^4\right)}{32}
\tag{11}
$$

As can be seen from Equations (9)–(11), only the profiles including component pairs 3 and 4 and 3 and 5 produce variance in the second moment of area (BSV) as a function of the phase angle $\phi$. However, Equation (11), in which the second moment of area is calculated for component pair 3 and 6, the phase angle $\phi$ does not affect the result. This demonstrates that the certain pairs do not produce the BSV. In Appendix B is more detailed discussion about BSV-producing component pairs.

Clearly, the profiles in Equations (9) and (10) produce BSV twice per revolution, as in the case of a single component in Equation (6), in which the profile included only the second harmonic component. The frequency in question can be seen in the equations as the sine function raised to the second power, or as the double angle in the sine function.

As can be noted from Equation (11) and Appendix B, not all component pairs produce BSV. The systematic pattern can be seen when the first 40 roundness components are studied in pairs similarly as in Equations (9)–(11) by observing phase angle $\phi$ in a final result. After all the pairs were studied, a $40 \times 40$ matrix could be constructed in which the BSV-producing pairs are indicated. This matrix can be seen in Figure 8.

Figure 8 shows that the BSV-producing component pairs form a systematic pattern. The component pairs that produce BSV can be divided into three different groups. In the first (green) group, the second components produce BSV regardless of the component pair. The components in the second group (blue) can produce BSV with the next two lower and higher components. In the third (orange) group, the BSV-producing component pairs are systematically scattered between two first groups. Obviously, the two same components at different phases do not produce BSV, as the diagonal in Figure 8 illustrates, except the second roundness component.

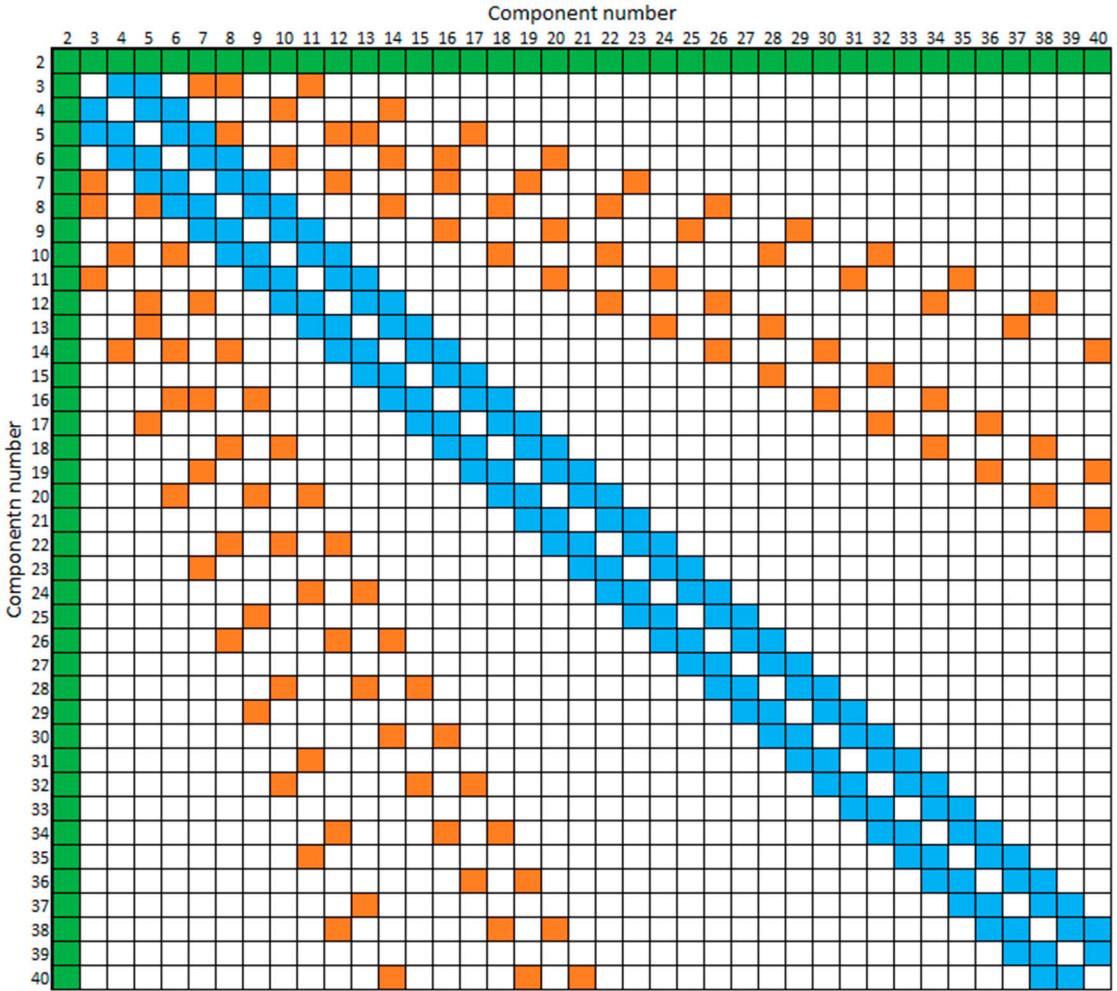

**Figure 8.** Correlation between component pairs with BSV. The pairs marked as white do not produce BSV, while the colored pairs produce. The different colors divide the components into three different groups; the green group includes the pairs containing the BSV-producing second component, the blue group includes the components that produce BSV with the next two lower and upper components, and the orange group includes the component pairs that are systematically scattered between the green and blue groups.

If the third component is interpreted in Figure 8, it produces BSV with the second, fourth, fifth, seventh, eighth, and 11th component, but not with the sixth, ninth, and 10th. Figure 9 presents these component pairs as profiles.

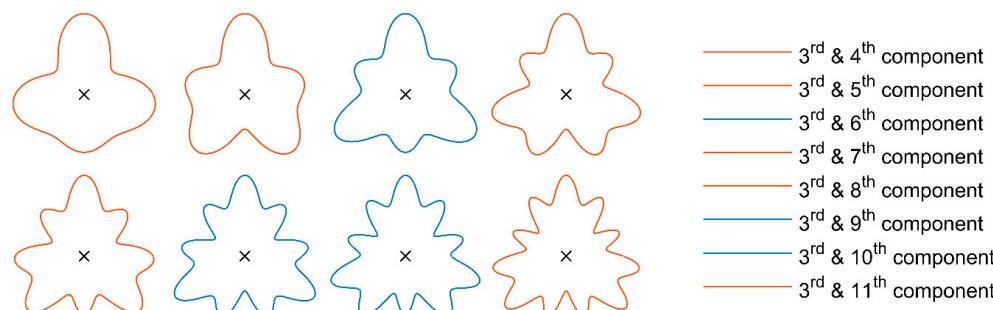

**Figure 9.** Profiles in which the third component is paired with components 4–11. The red profiles produce BSV and the blue profiles do not.

As can be seen from the profiles in Figure 9, it is impossible to state which profiles produce BSV, or which, in other words, have variation in the second moment of area as a function of the rotation angle.

The results demonstrate that the superposition principle, in which the effect of two or more components can be solved separately by combining the effects of single components, is not applicable when calculating BSV. As the results in Section 3.1 showed, only the second harmonic roundness component produced BSV when interpreting the effect of single harmonic roundness components. According to this and the superposition principle, components other than the second, or such component combinations, should not affect BSV. However, Equations (9) and (10) and Figure 8 indicate otherwise.

Equations (9)–(11) were solved with the shared phase angle, which rotated both components simultaneously. Hence, the formulas in that form cannot describe the effect of the component phases on BSV. Nevertheless, the effect of the component phase on BSV can be numerically studied and illustrated when BSV is calculated as a function of the component phase. In Figure 10, the BSV produced by component pairs 3 and 5 and 6 and 7 is illustrated in a 2D map as a function of the component phases. The radius of each calculated profile was 1 m, and the amplitude of each component was 0.1 m.

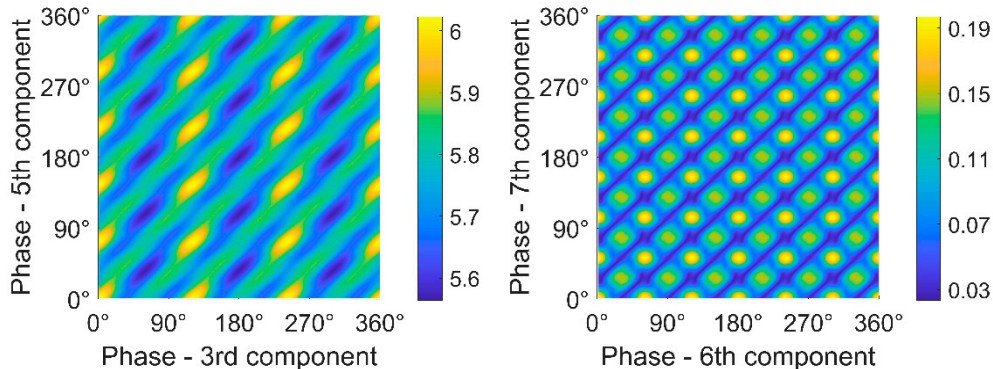

**Figure 10.** The effect of the roundness component phase on the BSV in a profile including two components.

Figure 10 demonstrates how the phase of the harmonic component affects BSV formation in a profile consisting of two harmonic roundness components. As the results show, the components have certain systematic phases that produce the maximal and minimal BSV. Obviously, these phases are divided according to the ordinal number of the harmonic component. Therefore, the rotation of the third component reaches its BSV maximum at 120° (360°/3) intervals, as Figure 10 shows.

The different component pairs seem to produce BSV very differently. In Figure 10, component pair 3 and 5 produces over 6% of BSV at maximum, while component pair 6 and 7 creates only a fraction of that, 0.2%. The relative phase of the components clearly affects the BSV produced. However, this variance is minor (<0.5%). Table 1 below calculates the maximum produced BSVs of component pairs 3–10. In the calculations, both component amplitudes were 0.1 m and the radius of the profile was 1 m. Hence, the maximum roundness error was 0.2 m at certain phase angles for each component pair. The selected amplitude for components was selected to be 10% of the radius of the profile, which corresponds to a realistic case.

**Table 1.** The maximum produced BSV [%] of component pairs 3–10. The pairs marked as white do not produce BSV, while the colored pairs produce. The different colors divide the components into three different groups; the green group includes the pairs containing the BSV-producing second component, the blue group includes the components that produce BSV with the next two lower and upper components, and the orange group includes the component pairs that are systematically scattered between green and blue groups.

| Harmonic Component | 2nd | 3rd | 4th | 5th | 6th | 7th | 8th | 9th | 10th |
|---|---|---|---|---|---|---|---|---|---|
| 2nd | 39.3 | 38.9 | 44.5 | 38.9 | 39.1 | 38.9 | 38.9 | 38.8 | 38.8 |
| 3rd | 38.9 | | 0.6 | 6.0 | | 0.3 | 0.6 | | |
| 4th | 44.5 | 0.6 | | 0.2 | 6.2 | | | | 0.5 |
| 5th | 38.9 | 6.0 | 0.2 | | 0.2 | 5.9 | 0.5 | | |
| 6th | 39.1 | | 6.2 | 0.2 | | 0.2 | 5.9 | | 0.5 |
| 7th | 38.9 | 0.3 | | 5.9 | 0.2 | | 0.2 | 5.9 | |
| 8th | 38.9 | 0.6 | | 0.5 | 5.9 | 0.2 | | 0.2 | |
| 9th | 38.8 | | | | | 5.9 | 0.2 | | 0.2 |
| 10th | 38.8 | | 0.5 | | 0.5 | | 5.9 | 0.2 | |

As Table 1 shows, the effect of the second component on BSV formation is markedly larger than for any other component. For all the pairs in which the second component is included (the green group), BSV is approximately 40%, while the other pairs reach only approximately 6% at maximum. By interpreting the pattern, one can notice that the 6% BSV is always produced by pairs formed with two higher or lower components.

The results in this section have shown that a roundness profile consisting of two certain roundness components can produce BSV that occurs twice per revolution. The formation of this kind of roundness profile, which includes or is dominated by two harmonic components, is possible and even common in industrial rotors that exhibit profiles mainly consisting of low-order harmonic components. According to the results, the significance of the second component is large; the pairs in which the second component was included produced at least 6–7 times larger BSV than the other pairs.

### 3.3. Bending Stiffness Variation of a Rotor Described with an Arbitrary Roundness Profile

An arbitrary roundness profile and the BSV produced can also be investigated, as presented in Section 2.1.3. Figure 11 depicts the arbitrary profile and its randomly deviated component amplitudes. Figure 12 illustrates the calculated second moment of area as a function of the rotation angle and its harmonic components. The profile has a radius of 1 m and it includes harmonic roundness components 3–50 with randomly generated amplitudes 0–0.1 m and phases 0–2π. The second component is eliminated, since it produced BSV also alone, as indicated in Section 3.1. The component amplitude distribution in the profile is focused on the lower end, which corresponds to typical manufacturing errors, in which the lowest components frequently dominate the roundness profiles of rotors.

The results show that the arbitrary profile produces BSV. The authors estimate that a part of the BSV is produced by certain component pairs, as discussed in Section 3.2, while the remainder is produced by more complex BSV-producing component combinations. Despite the fact that the profile was arbitrary and consisted of components from a wide range with different phases, Figure 12 shows that the BSV produced occurs twice per revolution, as in earlier results. This finding supports the assumption that there cannot be any other BSV excitation frequency than twice per revolution and the theory that profiles can have only two principal axes of second moment of area perpendicular to each other. The amplitudes of other components can be considered to be error caused by numerical calculation in Figure 12.

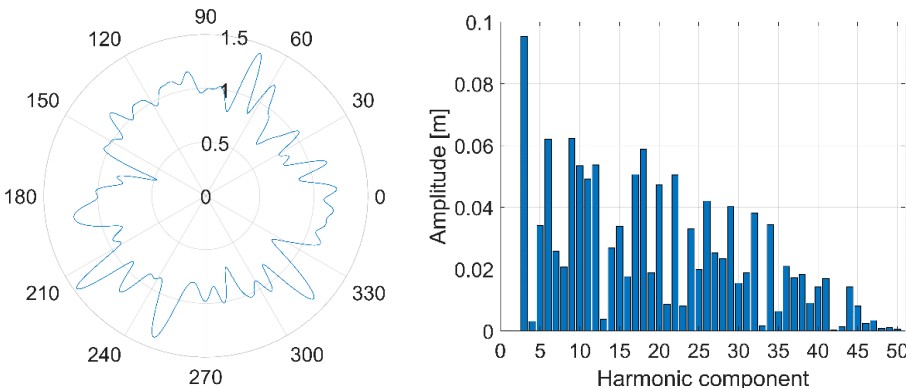

**Figure 11.** Generated solid profile including components 3–50 on the left-hand side. The component amplitude deviation is on the right-hand side.

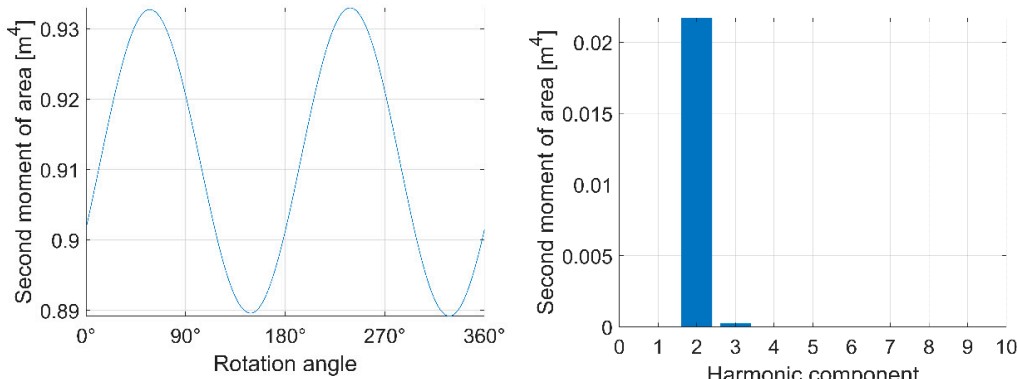

**Figure 12.** On the left-hand side, the variation of the second moment of area for generated solid profile including components 3–50. On the right-hand side, harmonic components of the variation.

### 3.4. Bending Stiffness Variation of a Rotor with a Varying Roundness Profile

Rotors with a varying roundness profile were investigated with the simulation explained in Section 2.2. Five different rotors were measured with different amounts of profile rotation. Figure 5 illustrates the four twisted rotor types that were measured; a rotor with a non-rotating roundness profile was used as the fifth reference rotor. This rotor type can be seen on the left-hand side of Figure 2. All the profiles included only the second component. In the simulation, the center point displacement of the rotor middle cross-section was acquired at 36 different loading angles, which led to 10° increments. The normalized results can be seen in Figure 13.

The normalized results were acquired with respect to the rotor with no rotation in the roundness profile, which yielded the largest BSV. Figure 13 clearly shows that BSV can be significantly decreased by distributing the BSV-producing roundness profiles along the rotor. BSV decreased in each rotor when the rotation was increased. Hence, it can be assumed that BSV approaches zero when the rotation angle approaches infinity.

Typically, the largest vibrations or rotor bending appear in the middle of the rotor. According to the results, the phase of the BSV-producing roundness profile can be used to compensate for BSV also in a single cross-section. This demands accurate positioning of the compensative roundness profiles along the rotor with certain amplitudes and phases. However, at other cross-sections, bending and vibration still occur, and thus BSV induced vibration cannot be eliminated completely with this method.

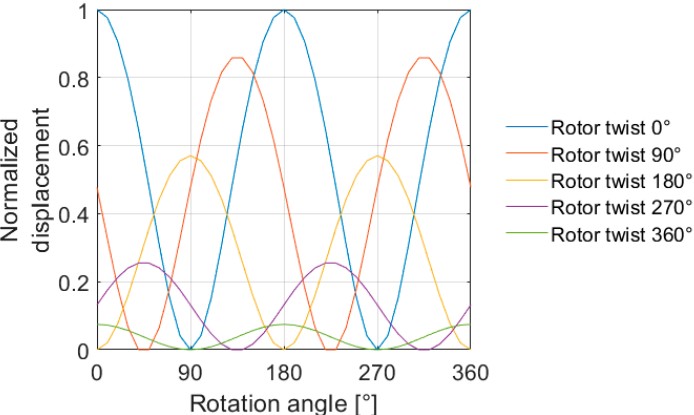

**Figure 13.** Normalized middle cross-section center point displacement of the rotor as a function of the rotational angle.

## 4. Conclusions

This study successfully investigated the effects of low-order harmonic roundness components on BSV, investigating separatelysingle components, pairs, and combination of a large number of arbitrary components. The broader investigation of different combinations was not considered necessary within the scope of this research, since rotors commonly feature a small number of dominant lower end components in their roundness profile after manufacturing. Moreover, machining devices repeatedly produce certain harmonic roundness components (manufacturing errors) in the roundness profile peculiar to the device.

The study presented novel analytical solutions for the effects of single components and component pairs on BSV. Solutions for more complex component combinations or rotor geometries must be done using numerical calculation or simulation. The results for single components demonstrated that only the second component was able to produce BSV; with other single components, the bending stiffness remained constant as a function of a rotation angle. Hence, any other single component than the second one, cannot affect the bending stiffness of a rotor. When component pairs were investigated, it was observed that only certain pairs produced BSV that was also possible without the second component. BSV amplitude comparison showed that the second component alone and pairs, in which the second component was included, produced significantly larger BSV amplitudes than other pairs. Therefore, the second component can be considered the most important BSV-producing component when attempting to eliminate BSV. Still, in extreme cases, certain component pairs can have large amplitudes that lead to serious vibration problems as well without the second component. Additionally, the study investigated arbitrary component combinations. It indicated clearly that BSV occurs always twice per revolution despite included components or their amplitudes and phases in a roundness profile. This shows that any rotor always has only two principal axes for second moment of area. In the final stage of this research, the scope was expanded to asymmetric rotors with a varying cross-section. The results showed that different BSV-producing cross-sections can compensate for each other attenuating overall BSV depending components phase. This enables new BSV manipulation methods in which rotor geometry is formed locally to achieve the desired compensation.

The study did not investigate more complex combinations analytically than component pairs, which leaves the possibility that there are other more complex component combinations which are capable to produce BSV. However, the authors estimate that the influence of these possible combinations are insignificant compared to the second component, especially when the component amplitudes typically diminish in higher components.

With presented methods and results, BSV problems due to rotor geometry can be investigated before a rotor delivery or alternatively when problems occur. Using the

derived results, the excessive BSV-producing component amplitudes can be recognized and handled afterwards. For a rotor prone to half critical vibration, the examination of manufacturing methods, rotor design, and rotor geometry is necessary. BSV is one potential reason for such vibration and is usually caused by rotor geometry errors. The findings of this paper to prevent BSV vibration can be used universally for different symmetric rotors in which BSV is caused by rotor geometry errors. The maximum amplitudes for each roundness component to limit BSV to a suitable level depend greatly on the rotor in question, and thus those must be determined separately for each rotor. Hence, still unknown matters such as the maximal limits of component amplitudes and their relation to rotor vibration are interesting research topics for the future.

**Author Contributions:** Conceptualization: R.V. (Risto Viitala); methodology: R.V. (Risto Viitala) and T.T.; validation: R.V. (Risto Viitala) and T.T.; formal analysis: R.V. (Risto Viitala); investigation: R.V. (Risto Viitala); writing—original draft preparation: R.V. (Risto Viitala); writing—review and editing: R.V. (Risto Viitala), T.T., and R.V. (Raine Viitala); visualization: R.V. (Risto Viitala); supervision: R.V. (Raine Viitala); project administration: R.V. (Risto Viitala) and R.V. (Raine Viitala); funding acquisition: R.V. (Risto Viitala) and R.V. (Raine Viitala) All authors have read and agreed to the published version of the manuscript.

**Funding:** This research was conducted during Digital Twin of Powertrain project, which was funded by Aalto University and ABB (Finland).

**Conflicts of Interest:** The authors declare no conflict of interest.

## Appendix A

Appendix A presents more detailed discussion about the effect of single harmonic roundness components on BSV. Focus is on BSV-producing components. The appendix is related to Section 3.1.

Variables used:

$k$ = Component number
$r$ = Function for the roundness profile (see Equation (4))
$A$ = Amplitude of the component
$\theta$ = Angle between the point on the profile and the horizontal axis
$\phi$ = Phase or rotation of the roundness profile

Roundness profile described with a single component in polar coordinates:

$$r = R + A sin(k(\theta + \phi)),$$

Second moment of area of a roundness profile $I_x$ described by the integral:

$$I_x = \int\limits_{0}^{2\pi} \int\limits_{0}^{r} r^3 \sin^2(\theta) dr d\theta$$

The variation of the second moment of area and bending stiffness can be investigated when the second moment of area $I_x$ is derived with respect to phase angle $\phi$. The derivative describes the variation of second moment of area and BSV as a function of profile rotation. Thus, when the derivative is zero with respect to roundness component, the component in question does not produce BSV. The derivative of second moment of area $I_x$ with respect to phase angle $\phi$ is presented below:

$$-\frac{1}{768k\left(36k^8-205k^6+273k^4-120k^2+16\right)}\left(-4608A^2kcos(2k\phi)R^2+4608A^3ksin(k\phi)R+6144Aksin(k\phi)R^3\right.$$
$$-1536ksin(3k\phi)RA^3-41472A^3k^7sin(k\phi)R-55296Ak^7sin(k\phi)R^3$$
$$+1536ksin(6\pi k+3k\phi)RA^3+4608A^2kcos(4\pi k+2k\phi)R^2+55296Ak^7sin(2\pi k+k\phi)R^3$$
$$-70272A^3k^5sin(2\pi k+k\phi)R-93696Ak^5sin(2\pi k+k\phi)R^3+33408A^3k^3sin(2\pi k+k\phi)R$$
$$+44544Ak^3sin(2\pi k+k\phi)R^3-1536k^7sin(6\pi k+3k\phi)RA^3+8064k^5sin(6\pi k+3k\phi)RA^3$$
$$-8064k^3sin(6\pi k+3k\phi)RA^3-10368A^2k^7cos(4\pi k+2k\phi)R^2-4608A^3ksin(2\pi k+k\phi)R$$
$$-6144Aksin(2\pi k+k\phi)R^3+70272A^3k^5sin(k\phi)R+93696Ak^5sin(k\phi)R^3$$
$$-33408A^3k^3sin(k\phi)R-44544Ak^3sin(k\phi)R^3+1536A^3k^7sin(3k\phi)R-8064A^3k^5sin(3k\phi)R$$
$$+8064A^3k^3sin(3k\phi)R+10368A^2k^7cos(2k\phi)R^2-48672A^2k^5cos(2k\phi)R^2$$
$$+29952A^2k^3cos(2k\phi)R^2+48672A^2k^5cos(4\pi k+2k\phi)R^2-29952A^2k^3cos(4\pi k+2k\phi)R^2$$
$$+41472A^3k^7sin(2\pi k+k\phi)R+8112A^4k^5cos(4\pi k+2k\phi)-8112A^4k^5cos(2k\phi)$$
$$+672k^3cos(8\pi k+4k\phi)A^4-108A^4k^7cos(4k\phi)+4992A^4k^3cos(2k\phi)-768A^4kcos(2k\phi)$$
$$+192kcos(4k\phi)A^4+1728A^4k^7cos(2k\phi)-672A^4k^3cos(4k\phi)+588A^4k^5cos(4k\phi)$$
$$-588k^5cos(8\pi k+4k\phi)A^4-1728A^4k^7cos(4\pi k+2k\phi)+768A^4kcos(4\pi k+2k\phi)$$
$$\left.-4992A^4k^3cos(4\pi k+2k\phi)+108k^7cos(8\pi k+4k\phi)A^4-192kcos(8\pi k+4k\phi)A^4\right)$$

when interpreting the roots of the denominator above, it can be observed that its roots match BSV-producing components. The positive integer roots of the denominator:

$$k_1=0,\quad k_2=1,\quad k_3=2$$

Because the number of the BSV-producing component must differ from zero, and the first component (eccentricity) cannot produce BSV in solid profiles, the second component is the only acceptable component from the solved roots. The result match with the symbolic calculation in Section 3.1.

The variation of second moment of area can also be numerically investigated by plotting it as a function of phase angle $\phi$ and component number $k$. However, amplitudes for the other variables $A$ and $R$ must be given before the plotting. The plots can be seen in Figure A1 below.

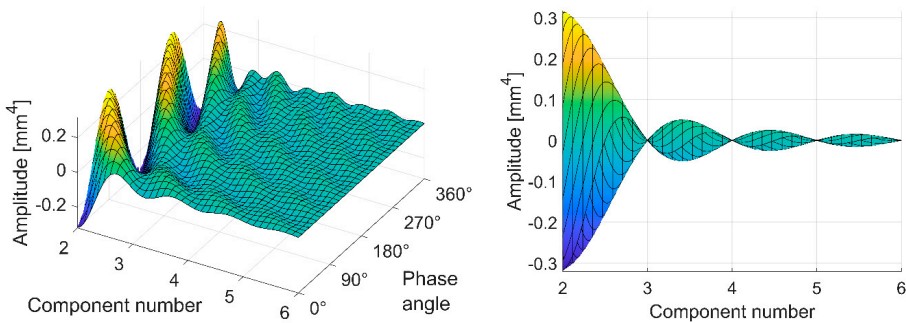

**Figure A1.** Derived second moment of area as a function of phase angle and component number. Both sides present the same plot at different angles. On the left-hand side can be seen the variation of second moment of area as a function of phase angle, and on the right-hand side can be seen the roots of the derived second moment of area with respect to component number.

As can be interpreted from Figure A1, only the second roundness component produces variation to second moment of area, since with other positive integer roundness components, the derivative is zero. In the left-hand side can be seen twice per revolution frequency with the second component. The same trend continues if more components are plotted. The first component (eccentricity) does not produce BSV in solid profiles, and therefore it is excluded from the results.

**Appendix B**

Appendix B presents more detailed discussion about the effect of harmonic roundness component pairs on BSV. Focus is on BSV-producing pairs. The appendix is related to Section 3.2.

Variables used:

$k_1$ = Component 1 number
$k_2$ = Component 2 number
$r(\theta)$ = Function for the roundness profile (see Equation (5))
$A_1$ = Amplitude of component 1
$A_2$ = Amplitude of component 2
$\theta$ = Angle between the point on the profile and the horizontal axis
$\phi$ = Phase or rotation of the profile

Roundness profile described with a component pair in polar coordinates:

$$r = R + A_1 sin(k_1(\theta + \phi)) + A_2 sin(k_2(\theta + \phi))$$

Second moment of area $I_x$ described by the integral:

$$I_x = \int\limits_0^{2\pi} \int\limits_0^r r^3 \sin^2(\theta) dr d\theta$$

BSV-producing pairs for a certain component can be calculated by solving the roots of the denominator of the second moment of area $I_x$ as in Appendix A. With this method, only the BSV-producing pairs for one component can be solved at a time. In this appendix, the pairs for the third component ($k_1$ = 3) are solved. Contrary to Appendix A, the definite integral for component pairs cannot be shown, as the solution is too large to reasonably present. Thus, only the denominator and its positive integer roots are presented:

$$574801920k_2{}^{24} - 166596756480k_2{}^{22} + 17795956590720k_2{}^{20} - 909008304807680k_2{}^{18}$$
$$+23946670539496320k_2{}^{16} - 332108916498670080k_2{}^{14} + 2415432619361435520k_2{}^{12}$$
$$-9220637695948250880k_2{}^{10} + 18002694347684027520k_2{}^{8} - 16837856981688314880k_2{}^{5}$$
$$+7158671392443648000k_2{}^{4} - 1284048424243200000k_2{}^{2} + 74798366720000000 = 0$$

$$\longrightarrow$$

$$k_2 = 2, \quad k_3 = 4, \quad k_4 = 5, \quad k_5 = 7, \quad k_6 = 8, \quad k_7 = 11$$

The roots correspond to the roundness components that produce BSV together with the third component, as also presented in Figure 8. Similar method can be also used to other roundness components to find BSV-producing pairs.

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
