# Peer review of "Effect of Rotor Geometry on Bending Stiffness Variation"

_machines, doi:10.3390/machines9020023_

Round 1
Reviewer 1 Report
The paper presents a very interesting discussion on the bending stiffness variation of rotors. That is definitely an interesting topic for the industry. The paper is very well written and organized.
The analytical results of sections 3.2 and 3.3 are the main contribution of the paper. This could be the basis of many diagnostics of rotors suffering from BSV a well as for corrective measures (as briefly mentioned in the paper).
The only minor suggestion is to clarify lines 17 to 22. For the first-time reader, it should be more clear what are the components mentioned.
Reviewer 2 Report
The paper investigates the effects of rotor geometry on the bending stiffness variation and further affecting the rotor vibration. Specifically, the geometry is influenced by the roundness. Numerical results are presented and discussions made. The topic is quite interesting. The following questions are raised for the authors to respond.
- Throughout the whole paper, reference to figures are not properly functioned as being displayed by ‘Error! Reference source not found.’ It causes much difficulty in interpreting the text. The authors are responsible when approving the submitted PDF.
- Researchers have conducted studies when the shaft cross section is irregular, i.e. not a circular one. Roundness as a relative new concept in rotordynamics, to the reviewer’s opinion, can be regarded machining errors and one type of irregular cross sectional shaft. How are they different in essence?
- It is said that the bending stiffness variation will be affected by rotor geometry as a function of rotational angle. This coincides with rotors with an open crack, see CNSNS, 83,2020, 105102. Therefore, can they share the same modeling method and dynamic behaviors? Please compare.
- In practical, we often see round shafts. Is this roundness really critical in operation? And how many numbers of components are we talking about for a normally produced rotor but with the supposed errors?
- For twisted shafts like the last ones in Figs.2-5, it still causes the BSV to be produced twice? I mean, dose the degree of roundness error have influence on the BSV pattern?
- The roundness can be a manufacture error, as pointed out in the paper. And the authors use randomly generated profile to simulate the shaft. This can be regarded geometrical uncertainties in the rotor system, for example JSV 466, 2020, 115047. The authors are suggested to strengthen the research background on this aspect.
- Please summarize your conclusion more concise to deliver the main and important findings. Readers may expect more direct impression from the conclusion what you have found.
- The references are too old except for those from the authors. If there is any more on relevant topics, please include them.
Reviewer 3 Report
The paper deals with the effect of geometrical characteristics of rotors' cross-section on its second moment of area. The paper contains too long introduction with certain missing information, the second section (Methods) introducing analytical solution of the second moment of area for given roundness expressions and certain general information on finite element models. The third section (Results and discussion) is devoted to the presentation of particular expressions for the second moment of area and the discussion of their dependancy on a phase angle, while subsection 3.4 contains certain short piece of the mechanical FEM analysis.
Without doubt, the topic of bending stiffness variation and its effects on the rotor characteristics is the interesting motivation and it is in the scope of the journal. The authors present detailed analysis of the second moment of area with respect to cross-sectional irregularities in their manuscript. In my opinion, the presented knowledge is quite limited and it does not bring new, utilizable and generalizable methods and conclusions. Moreover, the formal elaboration of the paper is poor. Therefore I do not recommend to consider this manuscript for the publication in the Machines journal and I suggest to the authors to substantially improve the manuscript and extend their work. Here are some particular points:
1) The analysis is mainly conducted for the second moment of area. Could you comment in more detail its relation with the stiffness?
2) Transition between Eq. 1 and Eq. 2 should be supported be a figure.
3) Eq. 3, Eq. 4 - are the brackets in "sin" arguments really correct? Should be the phase angle multiplied by "k"? If yes, can you explain it?
4) After substitution of Eq. 3 to Eq. 2, what will happen with differential "dr" and upper integral limit "r"?
5) I really wonder, how results presented by Eq. 5 to Eq. 7 were obtained. Please, include intermediate steps. Are really expressions for k = 3 and k = 4 the same?
6) How exactly was the matrix in Figure 6 obtained?
7) Results presented in Table 1 probably depend on the amplitude values of each components. Which values were considered in this analysis?
8) I recommend to add a figure for the definition of boundary conditions in case of FEM models.
9) The presentation of FEM results in 3.4 is very pure. Moreover, labels in Figure 11 are nor correct.
10) The presented research is pure theoretical work. In order to extend your research, try to establish more exactly defined methodology for the BSV elimination in the proper manufacturing phase. Or think about the extension to rotor dynamics problems.
11) The goal and the original contribution are missing in the introduction.
12) How "big" should BSV be to produce certain undesirable vibration in the real machines?
13) Several important formal issues:
The introduction is too "talkative" - it is tool long and it contains too many "however".
There are errors in automatic references in the whole paper!
Titles of subsection 2.1 and 2.2 are little bit confusing.
Missing brackets in Eq. 8 and Eq. 9.
Round 2
Reviewer 2 Report
The authors addressed majority of the comments. It is suggested to improve the comparison to previusly works of the authors themselves to highlight the difference.
Reviewer 3 Report
The authors somehow addressed all of my questions and notes. However, my main concern was not improved. I can repeat my sentence from the first review: In my opinion, the presented knowledge is quite limited and it does not bring new, utilizable and generalizable methods and conclusions.
In fact, what the authors did, is that they symbolically solved many expressions for the second moment of area and checked, whether it is dependent on the phase angle. Further, they accompanied it with the simple static finite element analysis for several geometrical types of rotors. It could be a nice research report or a student thesis. In order to extend your research for a scientific journal, I recommend trying to establish more exactly defined methodology for the BSV elimination in the proper manufacturing phase. Or think about the extension to rotor dynamics problems.
In this state, I prefer to leave the final decision for the journal editor. The paper is not wrong, and it contains corrects methods and findings. Just, it does not bring new knowledge in the scope suitable for a journal paper.
